# CONVERGENCE IS NOT ENOUGH: AVERAGE-CASE PERFORMANCE OF NO-REGRET LEARNING DYNAMICS

## ABSTRACT

Learning in games involves two main challenges, even in settings in which agents seek to coordinate: convergence to equilibria and selection of *good* equilibria. Unfortunately, solving the issue of convergence, which is the focus of state-of-the-art models, conveys little information about the quality of the equilibria that are eventually reached, often none at all. In this paper, we study a class of arbitrary-sized games in which *q-replicator* (QRD), a widely-studied class of no-regret learning dynamics that include gradient descent (GD), standard replicator dynamics (RD), and log-barrier dynamics as special cases, can be shown to converge *pointwise* to Nash equilibria. Turning to our main task, we provide both theoretical and experimental results on the *average case performance* of different learning dynamics in games. For example, in the case of GD, we show a tight average Price of Anarchy bound of 2, for a class of symmetric $2 \times 2$ potential games with unbounded Price of Anarchy (PoA). Furthermore, in the same class, we provide necessary and sufficient conditions so that GD outperforms RD in an average case analysis giving novel insights about two of the most widely applied dynamics in game theory. Finally, our experiments suggest that unbounded gaps between average case performance and PoA analysis are common, indicating a fertile area for future work.

## 1 INTRODUCTION

Multi-agent coordination often involves the solution of complex optimization problems. What makes these problems so hard, even when agents have common (Bard et al., 2020) or aligned interests (Dafoe et al., 2020; Dafoe et al., 2021), is that learning occurs on highly non-convex landscapes; thus, even if the learning dynamics equilibrate, their fixed points may include unnatural saddle points or even local minima of very poor performance (Dauphin et al., 2014). To address this issue, a large stream of recent work has focused on the convergence of optimization-driven (e.g., no-regret) learning dynamics to *good* limit points. Notable results include avoidance of saddle points and convergence of first order methods, e.g., *gradient descent*, to local optima (Ge et al., 2015; Lee et al., 2019; Mertikopoulos et al., 2019), point-wise or last-iterate convergence of various learning dynamics to (proper notions of) equilibria in zero-sum (competitive) games (Daskalakis & Panageas, 2019; Bailey & Piliouras, 2019; Cai et al., 2022) and convergence of no-regret learning to stable points in potential (cooperative) games ()HeliouCM17,PPP17,DBLP:journals/corr/abs-2203-12056,Leo22.

Even though these results seem to provide a sufficient starting point to reason about the *quality of the collective learning outcome*, unfortunately, this is far from being true. Non-trivial game settings routinely possess attracting points of vastly different performance, and this remains true, even if one is able to restrict attention to refined and highly robust notions of equilibria (Flokas et al., 2020).

Nevertheless, and despite the intense interest of the machine learning community to address the problem of *equilibrium selection*, there is a remarkable scarcity of work in this direction. To make matters worse, static, game-theoretic approaches to the problem (Harsanyi, 1973; Harsanyi & Selten, 1988; van Damme, 1987), offer little insight, often none at all, from a dynamic/learning perspective. In this case, the challenge is to show approximately optimal performance not for (almost) all initial conditions (which is not possible), but in expectation, i.e., for uniformly random chosen initial conditions (worst-case versus average-case analysis). This is a fundamentally hard problem since one has to couple the performance of equilibria to the relative size of their *regions of attraction*. However, regions of attraction are complex geometric manifolds that quickly become mathematically

intractable even in low-dimensional settings. Importantly, their analysis requires the combination of tools from machine learning, game theory, non-convex optimization and dynamical systems.

In terms of average case analysis of game theoretic dynamics in coordination/common interest games, the only other references that we know of are Zhang & Hofbauer (2015); Panageas & Piliouras (2016). In fact, Panageas & Piliouras (2016) is the key precursor to our work. Critically, whereas Panageas & Piliouras (2016) focuses exclusively on a single dynamics, i.e., replicator dynamics and bounding its average price of anarchy (APoA) in restricted instances of games such as Stag Hunt, we show how these techniques can be applied much more broadly by addressing novel challenges:

- **Axiomatic challenge:** Can we formally define the notion of Average Price of Anarchy for large classes of dynamics and games?
- **Analytical challenge:** Even if the definitions can be made robust how do we analyze these nonlinear dynamical systems given random initial conditions in the presence of multiple attractors?
- **Experimental/visualization challenge:** Can we develop novel custom visualization techniques as well as showcase that our experimental results have predictive power even in complex high dimensional settings?

**Model and Contributions.** To make progress in addressing these challenges, we study the *q-replicator dynamics* (QRD), one of the most fundamental and widely-studied classes of multi-agent learning dynamics that include *gradient descent*, *replicator* and *log-barrier* dynamics as special cases (A. Giannou, 2021). We start with our first motivating question which we answer affirmatively by proving *pointwise* convergence of all QRD dynamics to Nash equilibria (NEs) in almost all finite potential games. Potential games include multi-agent interactions in which coordination is desirable, congestion games and games of identical interests as important and widely-studied subclasses (Wang & Sandholm, 2002; Panait & Luke, 2005; Carroll et al., 2019; Dafoe et al., 2020).

The proof of point-wise convergence to NEs combines recent advances (Swenson et al., 2020)[1] with standard convergence techniques in the study of potential games, e.g., Palaiopanos et al. (2017b). Such techniques have been used to either establish convergence of QRD to NEs under the assumption of point-wise convergence (Mertikopoulos & Sandholm, 2016) or prove convergence to limit cycles of (restricted) equilibrium points (Mertikopoulos & Sandholm, 2018). However, whereas in previous works such results are the main focus, in our case they are only the starting point as they clearly not suffice to explain the disparity between the regularity of QRD in theory (bounded regret, convergence to Nash equilibria) and their conflicting performance in practice (agents' utilities after learning).

We then turn to our second question and the fundamental problem of equilibrium quality. While different QRD dynamics may reach the same asymptotically stable equilibria, this is only a minimal and definitely not sufficient condition to compare their performance. In particular, the *regions of attraction* of these common attracting equilibria, i.e., the sets of convergent initial conditions, can be very different for different QRD dynamics.

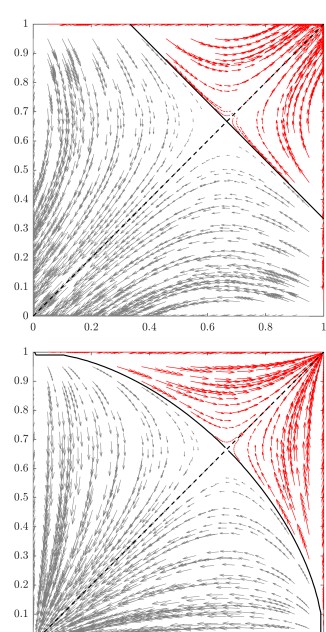

Figure 1: Vector fields of gradient descent (top) and replicator dynamics (bottom) for a game with payoff- and risk-dominant equilibrium at the bottom-left corner. The trajectories in the *region of attraction* of the good (bad) equilibrium are shown in gray (red). The black solid and dashed lines show the stable and unstable manifolds, respectively. In this case, gradient descent outperforms replicator dynamic.

---

[1] Specifically, Swenson et al. (2020) shows that all NEs in almost all potential games are regular in the sense of Harsanyi, i.e., they are isolated and highly robust Harsanyi (1973); van Damme (1987). *Almost all* refers to a set whose complement is a closed set with Lebesgue measure zero.

In our main technical contribution, we tackle this task by providing geometric insights into the shapes and sizes of the regions of attractions of different QRD dynamics. We show that in a class of two-agent potential games, gradient descent reaches the payoff-dominant (socially optimal) equilibrium more often than standard replicator whenever this equilibrium is also risk-dominant (less risky), see Figure 1. As an implication, we study a class of games in which the *Price of Anarchy* is unbounded, i.e., in which the worst-case equilibrium can be arbitrarily worse than the socially optimal outcome (Panageas & Piliouras, 2016), and derive a (tight) upper bound of 2 for the *Average Price of Anarchy* for the gradient descent dynamics and all instances of the class in which the risk- and payoff-dominant equilibria coincide. This is the first such tight result of its kind.

Conceptually, our methods provide a systematic approach to explore the design and hyperparameter space of learning dynamics and extend recent advances towards a taxonomy of learning dynamics in low-dimensional or general potential games (Panageas & Piliouras, 2016; Pangallo et al., 2022). More importantly, they signify the expressiveness in this task of performance measures that couple the likelihood of convergence to a certain outcome (region of attraction) with the performance of an algorithm at this outcome. From a practical perspective our findings admit a dual interperation. On the one hand, they provide concrete recommendations about the optimality of different QRD dynamics based on the features of the underlying game. On the other hand, they suggest, that even in the simplest possible classes of games, there is not a single optimal QRD dynamic to *beat them all*.

Intriguingly, the above results hinge on two interconnected, yet fundamentally different, theories. The first part (convergence), relies on the theory of Lyapunov analysis and the properties of dissipative systems, i.e., systems that lose momentum over time till they converge to a steady state.By contrast, the second part, i.e., the qualitative analysis of the different parametrizations of the QRD dynamics, relies on the existence of invariant functions that characterize stable and unstable areas in the state space of such systems (Palaiopanos et al., 2017a; Nagarajan et al., 2020). The existence of invariant functions, however, is a feature most often studied in conservative systems, a fundamentally orthogonal principle to the one of dissipation.

## 2 PRELIMINARIES: GAME-THEORETIC AND BEHAVIORAL MODELS

**Game-theoretic model.** A multi-agent finite potential game $\Gamma := \{\mathcal{N}, (\mathcal{A}_k, u_k)_{k \in \mathcal{N}}, \Phi\}$ denotes the interaction between a set $\mathcal{N} := \{1, \ldots, n\}$ of agents. Each agent $k \in \mathcal{N}$ has a finite set of actions, $\mathcal{A}_k$, with size $|\mathcal{A}_k|$, and a reward function $u_k : \mathcal{A} \to \mathbb{R}$ where $\mathcal{A} := \prod_{k \in \mathcal{N}} \mathcal{A}_k$ is the set of all *pure action profiles* of $\Gamma$. Agents may use mixed actions or *choice distributions*, $x_k = (x_{ka_k})_{a_k \in \mathcal{A}_k} \in \mathcal{X}_k$, where $x_{ka_k}$ is the probability with which agent $k$ uses their action $a_k \in \mathcal{A}_k$ and $\mathcal{X}_k := \{x_k \in \mathbb{R}^{|\mathcal{A}_k|} \mid \sum_{a_k \in \mathcal{A}_k} x_{ka_k} = 1, \ x_{ka_k} \geq 0\}$ is the $(|\mathcal{A}_k| - 1)$-dimensional simplex. Given any mixed-action $x_k \in \mathcal{X}_k$, we will write $\mathrm{supp}(x_k) := \{a_k \in \mathcal{A}_k \mid x_{ka_k} > 0\}$ to denote the *support* of the action $x_k$, i.e., the set of all pure actions $a_k \in \mathcal{A}_k$ that are selected with a positive probability at $x_k$. Using conventional notation, we write $s = (s_k, s_{-k}) \in \mathcal{A}$ and $x = (x_k, x_{-k}) \in \mathcal{X} := \prod_{k \in \mathcal{N}} \mathcal{X}_k$ to denote the *joint pure and mixed action profiles* of $\Gamma$, where $s_{-k}$ and $x_{-k}$ are the vectors of pure and mixed actions, respectively, of all agents other than $k$. When time is relevant, we will use the index $t$ for all the above, e.g., we will write $x_k(t)$ for agent $k$'s choice distribution at time $t \geq 0$. The function $\Phi : \mathcal{A} \to \mathbb{R}$ is called a *potential function* of $\Gamma$ and satisfies $u_k(s) - u_k(s'_k, s_{-k}) = \Phi(s) - \Phi(s'_i, s_{-i})$, for all $k \in \mathcal{N}$ and all $s, s' \in \mathcal{A}$. The agents' reward functions and the potential function extend naturally to mixed action profiles with $u_k(x) = \mathbb{E}_{s \sim x}[u_k(s)]$ and $\Phi(x) = \mathbb{E}_{s \sim x}[\Phi(s)]$.

**Regular Nash and restricted equilibria.** A *Nash equilibrium (NE)* of $\Gamma$ is an action profile $x^* \in \mathcal{X}$ such that $u_k(x^*) \geq u_k(x_k, x^*_{-k})$, for all $k \in \mathcal{N}$ and for all $x \in \mathcal{X}$. By linearity of expectation, the above definition is equivalent to:

$$u_k(x^*) \geq u_k(a_k, x^*_{-k}), \quad \text{for all } a_k \in \mathcal{A}_k, \text{ and all } k \in \mathcal{N}, \tag{1}$$

where $u_k(a_k, x^*_{-k})$ denotes the reward of agent $k$ when they play the pure action $a_k$, versus the mixed strategies $x^*_{-k}$ for the rest of the agents. Let $NE(\Gamma)$ denote the set of all NE of $\Gamma$. A NE is called symmetric if $x^*_1 = \ldots = x^*_n$, and is called *fully mixed* if $\mathrm{supp}(x^*) = \prod_{k \in \mathcal{N}} \mathrm{supp}(x^*_k) = \mathcal{A}$. A NE is called *regular* if it satisfies the following definition.

**Definition 2.1** (Regular Nash equilibria (Harsanyi, 1973; Swenson et al., 2020)). *A Nash equilbrium, $x^* \in NE(\Gamma)$, is called* regular *if it is (i) quasi-strict, i.e., if for each player $k \in \mathcal{N}$, $x^*_k$ assigns positive*

*probability to all best responses of player $k$ against $x^*_{-k}$* ~~all best responses of each player $k \in \mathcal{N}$ to~~ ~~$x^*_{-k}$ are contained in $x^*_k$~~*, and (ii) second-order non-degenerate, i.e., if the Hessian, $H(x^*)$, taken with respect to* $\mathrm{supp}(x^*)$ *is non-singular.*

Finally, a *restriction* of $\Gamma$ is a game $\Gamma' \coloneqq \{\mathcal{N}, (\mathcal{A}'_k, u'_k)_{k \in \mathcal{N}}\}$, where $\mathcal{A}'_k \subseteq \mathcal{A}_k$ and $u'_k : \mathcal{A}' \to \mathbb{R}$ is the restriction of $u_k$ to $\mathcal{A}' \coloneqq \prod_{k \in \mathcal{N}} \mathcal{A}'_k$ for all $k \in \mathcal{N}$. An action-profile $x \in \mathcal{X}$ is called a *restricted equilibrium* of $\Gamma$ if it is a Nash equilibrium of a restriction of $\Gamma$, cf. Mertikopoulos & Sandholm (2018). It is easy to see that all restrictions of a potential game $\Gamma \coloneqq \{\mathcal{N}, (\mathcal{A}_k, u_k)_{k \in \mathcal{N}}, \Phi\}$ are potential games, whose potential functions are restrictions of $\Phi$ to the respective subspaces of $\mathcal{A}$.

**Behavioral-learning model.** The evolution of the agents' choice distributions (or mixed actions) in the joint action space $\mathcal{X}$ is governed by the *q-replicator dynamics (QRD)* which are the parametric dynamics described by the system of differential equations (equations of motions) $\dot{x} \coloneqq V_q(x)$, where $V_q : \mathcal{X} \to \mathbb{R}^{|\mathcal{A}|}$ is given by:

$$\dot{x}_{ka_k} = x^q_{ka_k} \left( u_k(a_k, x_{-k}) - \frac{\sum_{a_j \in \mathcal{A}_k} x^q_{ka_j} u_k(a_j, x_{-k})}{\sum_{a_j \in \mathcal{A}_k} x^q_{ka_j}} \right), \quad \text{for all } k \in \mathcal{N}, a_k \in \mathcal{A}_k, \quad \text{(QRD)}$$

for any $q \geq 0$. Special cases of the above dynamics are the projection or *gradient descent (GD)* dynamics, for $q = 0$, the the *standard replicator (RD)* dynamics, for $q = 1$, and the *log-barrier* or inverse update dynamics, for $q = 2$ (Mertikopoulos & Sandholm, 2018; A. Giannou, 2021).

## 3 POINTWISE CONVERGENCE OF QRD TO NASH EQUILIBRIA

Our results consist of two parts. In the first part, which is the subject of this section, we show convergence of QRD to Nash equilibria in a class of potential games, which we term *perfectly-regular potential games*, whose definition follows.

**Definition 3.1** (Perfectly-regular potential games). *A potential game $\Gamma$ is called* regular *if it has only regular Nash equilibria. A regular potential game is called* perfectly-regular potential game (PRPG) *if all its restrictions are regular potential games, i.e., if they only possess regular Nash equilibria.*

Almost all potential games are PRPGs; this is a generalization of Swenson et al. (2020) who prove that almost all potential games are regular. Furthermore the PRPG class contains other important subclasses of games, e.g., congestion games, as well as games with identical reward functions, which are currently widely studied in the context of cooperative artificial intelligence (Wang & Sandholm, 2002; Panait & Luke, 2005; Carroll et al., 2019; Dafoe et al., 2020). The convergence result is stated formally in Theorem 3.2; its complete proof may be found in **??**.

**Theorem 3.2** (pointwise convergence of QRD to NE in PRPGs). *Given any perfectly-regular potential game (PRPG), $\Gamma$, and any interior initial condition $x(0) \in \mathrm{int}\, \mathcal{X}$, the q-replicator dynamics, defined as in equation QRD, converge pointwise to a Nash equilibrium $x^*$ of $\Gamma$ for any parameter $q \geq 0$. Furthermore, the set $\mathcal{Q}(\mathrm{int}\, \mathcal{X}) \coloneqq \bigcup_{x_0 \in \mathrm{int}\, \mathcal{X}} \{x^* \in \mathcal{X} \mid \lim_{t \to \infty} x(t) = x^*,\ x(0) = x_0\}$, i.e., the set of all limit points of interior initial conditions, is finite.*

*Sketch of the proof.* The proof of Theorem 3.2 proceeds in two steps, which utilize the properties that (i) PRPGs have a finite number of regular equilibria, and (ii) the probability of optimal actions *near* an equilibrium point is increasing in time with respect to the QRD. In the first step, we prove that for any initial condition, the sequence of joint action profiles $x(t)_{t \geq 0}$ that is generated by QRD for any $q \geq 0$ converges to a restricted equilibrium of a PRPG, $\Gamma$. This relies on the fact that the set of cluster (limit) points of the trajectory—also called the $\omega$-limit set—is a finite, and in fact, as we show, a singleton (a single element set) for any PRPG. In turn, this follows from the fact that a PRPG provably contains only a finite number of restricted equilibria.

Having established convergence to restricted equilibria, in the second step, it remains to show that, in fact, any such limit point has to be a NE of $\Gamma$, i.e., we need to exclude convergence to restricted equilibria that are not NE of $\Gamma$. To establish this, we couple the structure of PRPGs, which ensures that there is a finite number of (regular) restricted equilibria, with the nature of QRD which guarantees that in the vicinity of a limit point, optimal actions, i.e., best responses, need to be played with

increasingly higher probability. Thus, all actions in the support of the limit choice distribution of each agent must be best responses against the actions of all other agents, which implies that all points that can be reached by QRD are NE of $\Gamma$. $\square$

In other words, Theorem 3.2 says that for *almost all potentials games* and *almost all initial conditions*, QRD converge to a NE of the game. An important implication of Theorem 3.2 is that, when one is reasoning about the quality of the collective learning outcome in cooperative multi-agent settings (as captured by PRPGs), they can restrict their attention to NE. However, reverting to off-the-shelf, static performance measures that compare the quality of different NE, we only obtain results that are meaningless, if not misleading, from a dynamic/learning perspective. The reason is that certain *bad* (or sometimes even *good*) NE may be reachable only from a very small set of initial conditions. Thus, we need to develop and argue about *average* performance measures that couple the outcome of the learning process (NE of an PRPG), with the likelihood that such an outcome is reached by the given learning dynamic (region of attraction of this NE). This is the subject of the next section.

## 4 QUALITY OF THE COLLECTIVE LEARNING OUTCOME

**When static performance metrics fail.** Having established that in the landscape of potential games, QRD converge *almost surely* to Nash equilibria, we next turn our attention to the main challenge of quantifying the quality of the collective learning outcome. In order to do that, one would first have to establish *appropriate* performance metrics. In a static regime, we can rely in a variety of meaningful metrics, e.g., the *Price of Anarchy (PoA)* (Koutsoupias & Papadimitriou, 1999; Christodoulou & Koutsoupias, 2005; Roughgarden, 2015), which is defined as the ratio between the *socially worst* NE of the game and the *socially optimal* outcome, where the social-optimality of an outcome $x \in \mathcal{X}$ is measured with respect to the *social welfare* $\mathrm{SW}(x) := \sum_{k \in \mathcal{N}} u_k(x)$, i.e., the total reward of the agents. The PoA is a natural *static* metric that one may consider in a PRPG setup. After all, coordination is the essence of potential games, which typically model multi-agent settings where this is a desirable property. However, it is not difficult to find PRPGs where the PoA fails to provide any meaningful information about the game. Let us consider the following example:

**Example 4.1** (A simple example of unbounded performance loss). *Consider the parametric $2 \times 2$-PRPG, $\Gamma_w$, i.e., a 2-player 2-actions PRPG, with payoff functions $u_{w,1}(s_1, s_2) = u_{w,2}(s_2, s_1) = A_w(s_1, s_2)$, where the matrix $A_w \in \mathbb{R}^{2 \times 2}$ is given by:*

$$A_w = \begin{pmatrix} 1 & 0 \\ 0 & w \end{pmatrix}, \quad 1 \leq w. \tag{2}$$

*The games, $\Gamma_w$, are already expressive enough to capture the aforementioned problem. In order to see this, observe that the NE that corresponds to $x_1 = (1, 0)$ and $x_2 = (1, 0)$ has social welfare equal to $\mathrm{SW}(x) = 1 + 1 = 2$ but the NE that corresponds to $x'_1 = (0, 1)$ and $x'_2 = (0, 1)$ has $\mathrm{SW}(x') = w + w = 2w$. Since $w$ can take any value larger than 1, the difference in performance can be* arbitrary large *with respect to the PoA. Specifically, $PoA(\Gamma_w) = \frac{\mathrm{SW}(x')}{\mathrm{SW}(x)} = w \to \infty$ as $w \to \infty$.*[2]

### 4.1 REGIONS OF ATTRACTION AND AVERAGE PERFORMANCE MEASURES

While useful in static environments, the PoA metric fails to capture the dynamic nature of multi-agent learning. In particular, it does not provide an answer to the question:

> *How likely is it for the agents to reach a good or bad outcome given that the multi-agent system converges?*

To answer this question and argue about the collective performance of the game dynamics, we need to quantify the likelihood of each outcome when we the initial conditions of the system are randomly sampled. A region of attraction of a given outcome formalizes this notion.

---

[2]Note about notation: In a $2 \times 2$-game, one usually abuses notation and writes $x, y \in [0, 1]$ (instead of $(x, 1 - x)$ and $(y, 1 - y)$) to denote the mixed choice distributions of players 1 and 2, respectively. Then, all notions that we presented in Section 2 may be viewed as functions of $x, y$. For example, we could have written $\mathrm{SW}(1, 1)$ to denote the social welfare of the NE that corresponds to $x = y = 1$. In this section we are going to interchange between the two notations, but our choice is always going to be clear by the context.

**Definition 4.2** (Regions of attraction). *Let $\Gamma$ be any game and assume that its joint action profile, $x \in \mathcal{X}$, is evolving according to the equations of motion $\dot{x} = f(x)$. Then for any $x^* \in \mathcal{X}$, the set $RoA_{f,\Gamma}(x^*) := \{x_0 \in \mathcal{X} \mid \lim_{t \to \infty} x(t) = x^*, x(0) = x_0\}$ is called the* region of attraction (RoA) *of $x^*$ with respect to the dynamics $f$.*

In other words, the RoA of a point $x^* \in \mathcal{X}$ is the set of all initial conditions in $\mathcal{X}$ for which the dynamics asymptotically converge to $x^*$. Note that RoAs do not intersect. If we can determine the regions of attraction of some game dynamics, then given a certain static performance metric, e.g., the social welfare, we can define a corresponding *average-performance metric* that *weighs-in* all possible outcomes, in the sense of limit points, according to their likelihood of occurring with respect to the given dynamics. In order for this *average* to be *meaningful*, a minimum requirement, is that the dynamics converge for almost all, i.e., all but a measure zero, initial conditions. Formally, an average performance metric is defined as follows[3]:

**Definition 4.3** (Average-performance metric). *Let $\Gamma$ be a multi-agent game and assume that its joint action profile, $x \in \mathcal{X}$, is evolving according to the equations of motion $\dot{x} = f(x)$. Let $\mathcal{X}_0 \subseteq \mathcal{X}$ be a set of initial conditions such that the set of convergence points $\mathcal{Q}(\mathcal{X}_0)$ is finite. Then, given a performance metric $g : \mathcal{X} \to \mathbb{R}$ of $\Gamma$, the average-performance of the dynamics governed by $f$ in $\Gamma$ with respect to the performance metric $g$ and the set of initial condition $\mathcal{X}_0$, is given by*

$$APM_{g,\mathcal{X}_0}(f,\Gamma) := \sum_{x^* \in \mathcal{Q}(\mathcal{X}_0)} \mu(RoA_{f,\Gamma}(x^*)) \cdot g(x^*), \tag{APM}$$

*where $\mu$ is a probability measure on $\mathcal{X}_0$.*

In other words, an APM is the expected optimality of a random initialization of the dynamics in $\mathcal{X}_0 \subseteq \mathcal{X}$ with respect to some metric $g$. For instance, if the performance metric $g$ is the social welfare, then the average-performance metric with respect to $g$ measures the expected social welfare of the system for any random initialization in $\mathcal{X}_0$. The average-performance metric that we are going to use in the remainder of this section is the *Average Price of Anarchy (APoA)*. The APoA is an APM with respect to the social welfare, re-normalised such that the APoA is greater than equal to $1$, with equality only if (almost) all the initial conditions converge to the socially optimal outcome of the system. Formally, given a multi-agent game $\Gamma$, equations of motion $\dot{x} = f(x)$ that describe the evolution of the agents actions in $\Gamma$, and a set of initial conditions $\mathcal{X}_0 \subseteq \mathcal{X}$ that consists of almost all $\mathcal{X}$, the APoA is given by the formula:

$$\text{APoA}(f,\Gamma) = \frac{\max_{x \in \mathcal{X}} \text{SW}(x)}{\text{APM}_{\text{SW},\mathcal{X}_0}(f,\Gamma)}. \tag{APoA}$$

Here, it is important to note that Definition 4.3 does not ensure that an APM is always a *meaningful* metric for the system. However, as long as one can prove that (i) the dynamics converge pointwise to some $x^* \in \mathcal{Q}(\mathcal{X}) \subseteq \text{NE}(\Gamma)$ for almost all initial condition $x_0 \in \mathcal{X}$, and (ii) the set of limit points, $\mathcal{Q}(\mathcal{X})$, is finite —two conditions that are satisfied by any PRPG that evolves with respect to some QRD (cf. Theorem 3.2)—the APoA has an intuitive interpretation. Specifically, in this setup, the APoA is always bounded between the PoA and the *Price of Stability (PoS)* of the game, i.e., the ratio between the socially optimal outcome and the socially optimal NE.

## 4.2 THE TAXONOMY OF QRD IN $2 \times 2$ PRPGS

To systematically evaluate and compare the performance of different QRD in perfectly-regular finite potential games, we address the case of symmetric $2 \times 2$ coordination games, i.e., games in which one can change the identities of the players without changing the payoff to the actions. Such games constitute one of the current *frontiers* in terms of classification of game-dynamics (Zhang & Hofbauer, 2015; Pangallo et al., 2022). Such games are trivially potential games and include games of identical payoffs as special cases. Omitted definitions and proofs of this section may be found in **??**.

**Representation of symmetric $2 \times 2$ PRPGs.** Recall that a NE, $x^*$, of a symmetric potential $\Gamma$ is called *payoff-dominant* if $u_k(x^*) \geq u_k(x')$ for all $x' \in \text{NE}(\Gamma)$, and it is called *risk-dominant* if $x^*$ is

---

[3]For this definition, recall that a probability measure $\mu$ on a compact space $\mathcal{X}$ is a $\sigma$-additive function from the powerset of $\mathcal{X}$ to $\mathbb{R}_+$ such that $\mu(\mathcal{X}) = 1$ and $\mu(\mathcal{X}') \geq 0$ for all $\mathcal{X}' \subseteq \mathcal{X}$.

unilaterally optimal against the uniform distribution of the rest of the agents. All symmetric $2 \times 2$ PRPGs can be conveniently represented by the parametric class of games $\Gamma_{w,\beta}$, with payoff functions $u_{w,\beta,1}(s_1, s_2) = u_{w,\beta,2}(s_2, s_1) = A_{w,\beta,s_1,s_2}$, where the matrix $A_{w,\beta} \in \mathbb{R}^{2 \times 2}$ is given by:

$$A_{w,\beta} = \begin{pmatrix} 1 & 0 \\ \beta & w \end{pmatrix}, \quad \beta \leq 1 \leq w. \tag{3}$$

The game $\Gamma_{w,\beta}$ has the same NE as the original game, retains the payoff- and risk-dominance properties of its equilibrium points, and preserves the limiting behavior of any QRD (see **??**). Each game $\Gamma_{w,\beta}$ has three NE, two pure at $x = y = 0$ and $x = y = 1$, with social welfare $\mathrm{SW}(0,0) = 2w$ and $\mathrm{SW}(1,1) = 2$, respectively, as well as one fully-mixed NE at:

$$x^* = y^* = \alpha := \frac{w}{w + 1 - \beta}. \tag{4}$$

For convenience, we are going to refer to the first pure-NE as $x_w$. Note that $x_w$ is payoff-dominant for any parametrization $\Gamma_{w,\beta}$, and it is also risk-dominant whenever $w > 1 - \beta$, or equivalently, whenever $\alpha > 0.5$. The first result of this section states that whenever the risk- and payoff-dominant equilibria of $\Gamma_{w,\beta}$ coincide, i.e., $\alpha \geq 0.5$, then the gradient descent dynamics, i.e., the 0-replicator dynamics, perform better (or equally in the generic case $\alpha = 0.5$) on average than the standard replicator dynamics with respect to the social welfare of their outcomes, i.e., they yield a smaller APoA. In any other instance of these games, i.e., for $\alpha < 0.5$, the RD perform better than GD with respect to the same metric.

**Theorem 4.4** (Performance of QRD in symmetric $2 \times 2$ PRPG). *Given any $2 \times 2$ symmetric PRPG, which, without any loss of generality, can be represented as an instance $\Gamma_{w,\beta}$, it holds that*

$$APM_{\mathrm{SW,int}} \chi(V_0, \Gamma_{w,\beta}) \geq APM_{\mathrm{SW,int}} \chi(V_1, \Gamma_{w,\beta}) \tag{5}$$

*if and only if ~~whenever~~ the payoff-dominant equilibrium is also risk-dominant, with equality only when ~~if and only if~~ $\alpha = 0.5$, i.e., $w = 1 - \beta$, where $V_0, V_1$ are the equations of motion of the 0-replicator and 1-replicator dynamics, respectively ~~equation QRD~~.*

**Interpretation of Theorem 4.4.** The proof of Theorem 4.4 proceeds with a first order analysis of the manifolds that separate the regions of attractions of the two pure equilibria for the different dynamics (cf. Figures 2 and 3). When comparing the gradient descent (GD) dynamics and the replicator dynamics (RD), the main implication of this theorem is that the expected social welfare is optimized by GD whenever risk and payoff-dominant equilibria coincide and is optimized by RD when risk and payoff-dominant equilibria differ. More generally, this result may be interpreted in two ways. On the one hand, it provides a concrete recommendation on the optimal behavior of the agents (GD versus RD) based solely on the properties of the underlying game. On the other hand, it suggests that even in the low-dimensional setting of $2 \times 2$ potential games, there is not a uniform recommendation, and the optimal behavior largely depends on the features of the underlying game. As it turns out, in this case, the decisive feature is the *riskiness* of the payoff-dominant equilibrium.

**Generalization to all QRD.** Technically, the proof of Theorem 4.4 uses tools that are orthogonal to the Lyapunov analysis, and the theory of dissipation of dynamical systems, that we used to prove convergence to NE in section 3. It leverages the *constants of motion* or *invariant functions* (Nagarajan et al., 2020), i.e., quantities that remain constant along the trajectories of the learning dynamics. The rationale is that if one could identify such a function, then, by finding its value at the unique mixed equilibrium $\alpha$ of the game, they can determine all initial conditions that asymptotically converge to it: these will be all points at the same level set of the invariant function. The manifold, i.e., the geometric locus, of all the points that converge to the equilibrium, i.e., the *stable manifold* of $\alpha$, is the one that separates the regions of attractions of the two pure NE of the game. Because of this property, we may also refer to the stable manifold of the mixed NE as the *separatrix* (Panageas & Piliouras, 2016). Note that, since the dynamics are also backward-invariant (Panageas & Piliouras, 2016; Mertikopoulos & Sandholm, 2018), their level-set will also contain a set of initial conditions that converge to it when moving backward in time. This points constitute the *unstable manifold* of $\alpha$. In the following lemma we identify such an invariant for all QRD.

**Lemma 4.5** (Invariant functions of QRD in $2 \times 2$ symmetric PRPGs). *Given a $2 \times 2$ symmetric PRPG, $\Gamma_{w,\beta}$, whose agents evolve with respect to the q-replicator dynamics, the separable function $\Psi_q : (0,1)^2 \to \mathbb{R}$ with $\Psi_q(x, y) := \psi_q(x) - \psi_q(y)$, where $\psi_q : (0,1) \to \mathbb{R}$ is given by:*

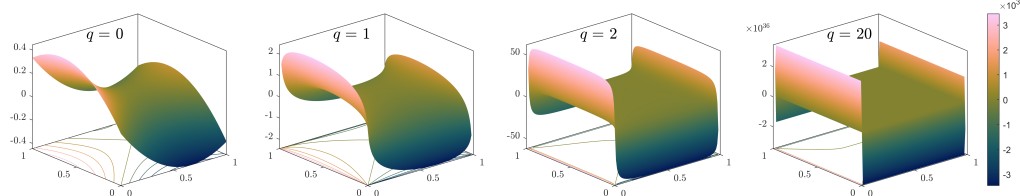

Figure 2: The invariant function, $\Psi_q(x, y)$, for all $x, y \in [0, 1]^2$ in the game $\Gamma_{w,\beta}$ for $w = 2$, $\beta = 0$, and various values of $q$: $q = 0$ (gradient descent), $q = 1$ (standard replicator), $q = 2$ (log-barrier), and $q = 20$. The invariant function becomes very steep at the boundary as $q$ increases, taking both arbitrarily large negative (**dark**) and positive (light) values in the vicinity of the NE.

$$\psi_q(x) = \begin{cases} \dfrac{x^{2-q} + (1-x)^{2-q} - 1}{2-q} + \dfrac{1 - \alpha x^{1-q} - (1-\alpha)(1-x)^{1-q}}{1-q}, & q \neq 1, 2, \\[2mm] \alpha \ln(x) + (1-\alpha)\ln(1-x), & q = 1, \\[2mm] \ln(x) + \ln(1-x) + \dfrac{\alpha}{x} + \dfrac{1-\alpha}{1-x}, & q = 2, \end{cases} \qquad (6)$$

*remains constant along any trajectory $\{x(t), y(t)\}_{t \geq 0}$ of the system. The function $\Psi_q(x)$ is continuous with respect to the parameter $q$ at, both, $q = 1$ and $q = 2$, since $\lim_{q \to 1} \Psi_q(x) = \Psi_1(x)$ and $\lim_{q \to 2} \Psi_q(x) = \Psi_2(x)$ for all $x \in (0, 1)$.*

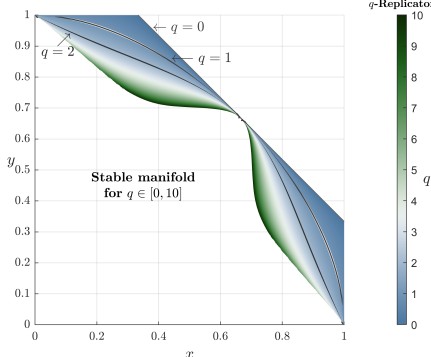

Figure 4: Stable manifold (separatrix) for all different values of $q \in [0, 10]$ (from blue to brown) in the $\Gamma_{w,\beta}$ game for $w = 2$ and $\beta = 0$. The manifolds for $q = 0$, $q = 1$, and $q = 2$ are shown in shades of black for reference (cf. Figure 3). The region of attraction of the payoff-dominant equilibrium (bottom-left corner) shrinks as $q$ increases.

In Figure 2, we visualize the invariant function, $\Psi_q(x, y)$, for $x, y \in (0, 1)^2$ for various values of $q \in [0, 20]$. From the panels of Figure 2, it is also evident that $\Psi_q(x, y)$ acts as a handy tool to visualize the regions of attraction of the two pure NE of the game. Namely, at the unique mixed NE, i.e., at $x = y = \alpha$, the invariant function, $\Psi_q$, is equal to 0. The same holds for any point $(x, y) \in (0, 1)^2$ with $x = y$. Thus, we can factorize $\Psi_q(x, y)$ as $\Psi_q(x, y) = \Psi_{q,\text{Stable}}(x, y) \cdot (x - y)$ where $\Psi_{q,\text{Stable}}(x, y) = 0$ is precisely the geometric locus of all points $(x, y) \in (0, 1)^2$ such that $\lim_{t \to \infty} x(t) = \alpha$, and $y = x$ is the geometric locus of all points such $\lim_{t \to -\infty} x(t) = \alpha$. These two manifolds constitute the *stable* and *unstable* manifolds, respectively, of the $q$-replicator dynamics.

Since the invariant function $\Psi_q(x, y)$ takes the value 0 only at the stable and unstable manifolds, we can visualize the separatrix for different values of $q$ by plotting the 0-level set of the invariant functions in Figure 2. These are depicted in Figure 3. As a sanity check, we also see from Figure 3 that the region of attraction of the payoff-dominant equilibrium for $q = 0$ (GD dynamics) is larger than the region of attraction for $q = 1$ (RD).

**Empirical evidence for the monotonicity of the APM with respect to $q$.** If we stack the stable manifolds (solid blue lines) in the panels of Figure 3, it becomes evident that the region of attraction of the payoff-dominant and risk-dominant equilibrium grows as $q$ decreases to 0. This is depicted in Figure 4 for all values of $q \in [0, 10]$ (the progression of the surface remains essentially unchanged for larger $q$). Analogous plots (but with the results reversed as predicted by Theorem 4.4) can be generated for instances of $\Gamma_{w,\beta}$, in which the risk-dominant equilibrium is different from the payoff-dominant one, as well as, for $2 \times 2$ generic PRPGs (cf. section 4). In general, putting together Theorem 4.4 and the aforementioned visualizations, we have both *theoretical* and *empirical* evidence that the region of attraction of the payoff-dominant equilibrium in $\Gamma_{w,\beta}$ is decreasing (increasing) in $q$ for $q \geq 0$ whenever this equilibrium is (is not) risk-dominant. Formal verification of the monotonicity of

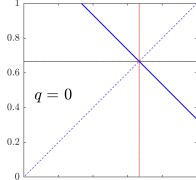 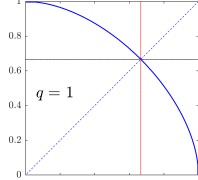 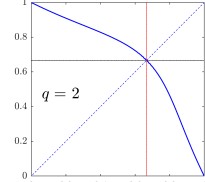 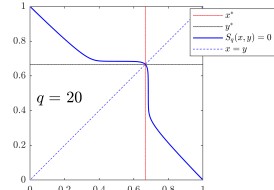

Figure 3: The stable manifolds, $\Psi_{q,\text{Stable}}(x, y) = 0$, (solid blue lines) for the same values of $q$ and the same instance of $\Gamma_{w,\beta}$ as in Figure 2, in which the payoff- and risk- dominant NE is at the bottom left corner. For all $q$, the separatrix goes through the mixed NE at the intersection of the $x^*$ (dashed red) and $y^*$ (dashed black) coordinates. All panels also include the unstable manifold defined by $x - y = 0$ (dashed blue line). The region of attraction of the payoff-dominant NE is larger for all values of $q$; however, this is because this NE is also risk-dominant, cf. Theorem 4.4.

~~the stable manifolds~~ the regions of attractions with respect to $q$ in the QRD parametrization remains open.

**Application: APoA in $2 \times 2$ PRPGs.** We conclude this section by providing a concrete result regarding the evaluation of the APoA average-performance measure in the class of $2 \times 2$ symmetric PRPGs, which showcases the practical importance of Theorem 4.4 and the invariant function approach.

**Theorem 4.6.** *The APoA of GD dynamics in all $2 \times 2$ symmetric PRPGs, $\Gamma_{w,\beta}$, is bounded by 2, i.e., $APoA(V_0, \Gamma_{w,\beta}) \leq 2$. Furthermore, this bound is tight.*

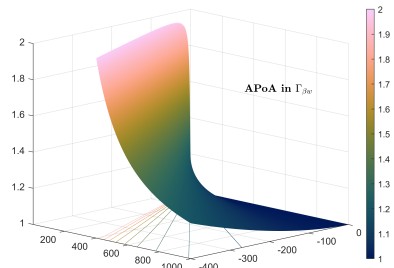

Figure 5: APoA of a $2 \times 2$ symmetric PRPG for the gradient descent dynamics and various values of $\beta$ and $w$. The APoA is upper bounded by 2 (**dark** to light values) as shown in Theorem 4.6.

The bound also holds for $\beta = 1 - w$, but in this case, there exists no risk-dominant equilibrium. The proof of Theorem 4.6 essentially proceeds by first order analysis of the function depicted in Figure 5 which, in turn, depends on the invariant function of the gradient descent dynamic. One way to see that this bound is tight, is to set $\beta = 1 - w + \epsilon$, for a small $\epsilon > 0$ and let $w$ increase (cf. Figure 5). In combination Theorem 4.4 and Theorem 4.6 imply that the APoA of the RD (QRD with $q = 1$), is *not* upper bounded by 2 whenever $\alpha < 0.5$, i.e., whenever the risk- and payoff-dominant equilibria are different. However, for the case $\alpha > 0.5$, the separatrices for all $q \geq 0$ as visualized in Figure 4, (empirically) imply that similar bounds hold for all values of $q$.[4] In **??**, we run simulations of $q$-replicator dynamics which provide evidence that the statement of Theorem 4.4 and the bound of Theorem 4.6 continue to hold in PRPGs of higher dimensions, i.e., beyond the $2 \times 2$ setting.

## 5 CONCLUSIONS

In this paper, we studied the class of $q$-replicator dynamics (QRD), and showed that all QRD converge pointwise to Nash equilibria in perfectly-regular potential games, a class of games that encompasses almost all potential games, i.e., the standard models of multi-agent coordination. The convergence of QRD in these settings is remarkably robust, occurring regardless of the number of agents or actions and for all possible parametrizations of QRD. From the perspective of equilibrium selection and quality, however, convergence provides little information, often none at all. Turning to this challenging problem, we provided geometric insights into the reasons why different dynamics exhibit fundamentally different performance despite their convergence to the very same set of attracting points. Our techniques leverage two intertwined, yet orthogonal to each other elements of dynamical systems theory: dissipation (Lyapunov theory) and conservation (invariant functions).

---

[4]To avoid confusion, in Figure 4, we visualize the stable manifolds for the case in which GD are the dynamics with the *largest* region of attraction, i.e., has the lowest APoA. The case $\alpha < 0.5$, in which the manifolds are simply mirrored on the $y = 1 - x$ diagonal, is in **??**.

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
