# OpenReview forum: "Convergence is Not Enough: Average-Case Performance of No-Regret Learning Dynamics"
_ICLR.cc/2023/Conference — Submitted to ICLR 2023_

### Official Review · Reviewer_jh9A · 2022-10-27

**Confidence:** 4
**Clarity, Quality, Novelty And Reproducibility:** Very well written paper.
**Correctness:** 4
**Technical Novelty And Significance:** 4
**Empirical Novelty And Significance:** 2
**Recommendation:** 8

**Strength And Weaknesses:**

Strengths
======================
  - The problem that the authors consider is very important and difficult and even the results for the special case of 2-by-2 games are very important.

  - The paper is well-written and the algorithms, techniques, and results are well-explained.

Weaknesses - Comments
========================

  1. 2-by-2 games is a rather restricted case, it would be nice if the authors can explain why their techniques cannot be extended to more general games. Do we have lower bounds for more general games.

Post-Rebuttal
========================
I thank the authors for the response! After discussion with other reviewers, I agree in the following weaknesses of the paper:
a. The contribution of the paper should be expressed more clearly. In particular, it should be clear from the abstract that this is not the first work that explores the average case performance of no-regret algorithms in games. Also, I believe the authors should highlight more, even in the abstract, that the class of games that they consider captures games for which the worst-case PoA is unbounded.
b. The number of people that would be interested in this result is smaller compared to other conferences, e.g., EC, which are a better fit for this paper.
I still recommend acceptance but I would reduce my score to 7 instead of 8 if there was such an option.

**Summary Of The Paper:**

In this paper, the authors consider the problem of finding dynamics that not only converge to Nash Equilibria but also converge to "good" equilibria according to some objective function, e.g., the social-welfare. The main results are the following:
  1. The social welfare of the 0-replicator dynamics is higher than the social welfare of the 1-replicator dynamics.
  2. For the special case of symmetric two player game with two strategies (2-by-2 games) the price-of-anarchy, i.e., the ratio of the optimal social welfate divided with the social welfare of the equilibrium given by the dynamics, is bounded by 2.

**Summary Of The Review:**

  Based on my comments above I recommend acceptance.

---

> ### Author Response · Authors · 2022-11-16
> **Re: Official Review of Paper6539 by Reviewer jh9A**
>
> We thank the reviewer for the feedback and we are glad that the reviewer appreciated the paper.
> It is an interesting open question to extend our findings to more general games. To be able to
> theoretically analyze more general games, one should be able to characterize the regions of attraction
> of each attracting equilibrium. Such a task is mathematically challenging as it is not guaranteed that
> we have closed form solutions for the boundaries of the regions of attraction (manifolds). It is also
> computationally challenging, as computing the regions of attraction for general games is NP-Hard.
> Nevertheless, it is quite easy to have an empirical analysis: for multiple random initializations one
> can count the frequencies for each limiting equilibrium and approximate the average price of anarchy.

---

### Official Review · Reviewer_S7pu · 2022-10-28

**Confidence:** 3
**Correctness:** 4
**Technical Novelty And Significance:** 2
**Empirical Novelty And Significance:** 2
**Recommendation:** 5

**Clarity, Quality, Novelty And Reproducibility:**

The paper is clearly written.
The problem is of interest to the community.

Additional comments:
- Theorem 4.4, as stated, does not say anything about the case $\alpha < 0.5$, but the preceding paragraph seems to suggest that when $\alpha \geq 0.5$, RD is better than GD. Can you clarify which is correct, the theorem or the paragraph?
- Please avoid overloading notation. $\alpha$ is used both in equation (4) and later to denote an equilibrium of the game.
- On page 8, the factorization of $\Psi_q(x, y)$ should be explained better.
- I find the abstract to be an inaccurate account of what is actually done in the paper. It should be mentioned at least that the main results concern 2x2 symmetric coordination games.
- The formatting of citations should be improved. In its current form, it hurts readability.
- Top of page 9: "monotonicity of the stable manifolds": did you mean of the regions of attraction?

**Strength And Weaknesses:**

I found the paper to be well written and easy to read. It is certainly convincing in terms of motivation, the problem is interesting and challenging.

My main concern is that the paper asks interesting questions, but makes little progress in terms of answering them. The paper sets out to compare the average performance of q-replicator dynamics for the class of symmetric 2x2 coordination games. Even in this 2-dimensional case, the results presented in the paper are very partial and feel preliminary in nature:
- Theorem 4.4 compares average performance between q=0 and q=1, when $\alpha \geq 0.5$. Nothing is said about other values of $q$.
- An invariant function is given in Lemma 4.5, which gives empirical evidence (through plots) that the regions of attraction are monotonic in q, but no formal proof is given. This left me with the impression that there is no treatment of the general case that was promised in the introduction (i.e. general q) except a conjecture at this point.
- It is also unclear if/how this geometric approach would generalize to other game classes, especially in higher dimensions. It would be great if the authors could argue why this approach can generalize and can be useful beyond the two dimensional case.


**Summary Of The Paper:**

The paper motivates the study of the relative sizes of regions of attraction of different equilibria, then tackles the case of symmetric, 2x2 coordination games. For this class of games, some partial/preliminary results are given.

**Summary Of The Review:**

The problem is relevant and the empirical/visual approach is promising. This can make a fine contribution if the formal results are expanded to have more formal proofs or a more complete treatment of the 2x2 symmetric games case.

---

> ### Author Response · Authors · 2022-11-16
> **Re: Official Review of Paper6539 by Reviewer S7pu**
>
> We thank the reviewer for their feedback. Below we address the concerns.
>
> >- "*the paper asks interesting questions, but makes little progress in terms of answering them*"
>
> The scope of the paper is to initiate the analysis of the performance of no-regret algorithms that are guaranteed to converge to equilibrium solutions. Apart from our theoretical and experimental results, we firmly believe that initiating this line of research is a contribution of this paper. In the literature, the community of ”learning in games” has mainly focused on providing algorithms with convergence guarantees but the works that address the quality of the solutions beyond worst-case Price-of-Anarchy analysis are very few; as opposed to the optimization community in which people have focused on proving saddle point avoidance and convergence to second order stationarity (better quality solutions). We firmly believe that initiating this line of work will attract a lot of interest in the near future.
>
> >- "*Theorem 4.4, as stated, does not say anything about the case $\alpha < 0.5$, but the preceding paragraph seems to suggest that when $\alpha \geq 0.5$, RD is better than GD*"
>
> Theorem 4.4 is an if and only if statement, so it says what happens both when $\alpha \geq 0.5$ and when $\alpha < 0.5$. We slightly rephrased Theorem 4.4. to indicate this. So, both the Theorem and the previous 1 paragraph are correct.
>
> >- "*overloading notation*"
>
> $\alpha$ denotes throughout the equilibrium of the game. So $\alpha$ in equation (4) is the same as $\alpha$ later.
>
> >- "factorization of $\Psi_q(x,y)$ should be explained better"
>
> The factorization of $\Psi_q$ is based on the discussion of page 8: $(x − y)$ is a factor (a zero) of function $\Psi$.
>
> >- "It should be mentioned at least that the main results concern 2x2 symmetric coordination games"
>
> We changed the abstract to clarify that we show point-wise convergence to Nash equilibria for
> games of general size (not only 2x2). For 2x2 games we can do average price of anarchy analysis for
> q-replicator dynamics.
>
> >- "The formatting of citations should be improved"
>
> We agree with the reviewer’s comment, but we have used the standard ICLR format. We are happy
> to change it if we are allowed.
>
> >- "Top of page 9: "monotonicity of the stable manifolds": did you mean of the regions of attraction?"
>
> We fixed it in the write-up, it is the same.

---

> > ### Comment · Reviewer_S7pu · 2022-11-16
> > **Re: Official Review of Paper6539 by Reviewer S7pu**
> >
> > Thank you for the response and the updates.
> >
> > Point taken about initiating the study of average-case performance, though one could argue that this was already initiated by prior works you cite, such as PP16 (albeit in more restricted settings). When I read the abstract (a statement such as "we quantify both conceptually and experimentally the outcome of optimal learning dynamics"), I was expecting a more systematic treatment of the question, but the formal results remain partial at this point.
> > I certainly think this is a question worth investigating, and I will take this point under consideration when discussing the paper with other reviewers.
> >
> > Some minor clarifications:
> > - The earlier statement of Theorem 4.4 used "whenever", which usually means a necessary condition, not the same as "if and only if". So this is quite an important distinction.
> > - About citations: by improving formatting, I was alluding to a better use of \citet, \citep, etc., within the standard ICLR format. For example (from the introduction), "even when agents have common Bard et al. (2020) or aligned interests" would read much better as "even when agents have common (Bard et al. 2020) or aligned interests...".
> > - I don't think monotonicity of stable manifolds is the same as monotonicity of regions of attraction. Monotonicity of sets is usually defined in terms of set inclusion, $S_q$ is monotone increasing means that $S_q \subseteq S_{q'}$ if $q \leq q'$. This makes sense for regions of attraction, but I don't see how it applies to stable manifolds.

---

> > > ### Author Response · Authors · 2022-11-16
> > > **Response to Reviewer S7pu**
> > >
> > > We thank Reviewer S7pu for considering our response and for their quick, additional clarifications. They are well-received: in our newest upload,
> > > 1. we modified the abstract to accurately describe our contributions,
> > > 2. we clarified the text in and before Theorem 4.4 (the proof in the appendix remained unaltered since it was for the correctly intended "if and only if" statement),
> > > 3. we fixed the format of the references, and
> > > 4. we removed the confusing statement about the monotonicity of the manifolds.
> > >
> > > Concnerning the latter, let us clarify why we suggested that they are the same in this case, but not in general, as correctly pointed out by the reviewer. Figure 3 shows that the manifolds of these games can be interpreted as $x-y$ functions. Thus, their monotonicity, i.e., for each $x$, the $y$ value increases as $q$ decreases, can be directly associated with monotonicity of the regions of attraction. Based on this visual evidence which is summarized in Figures 4 and 8, we formulated the open question of proving monotonicity of the regions of attraction by proving monotonicity of these functions. We hope that this clarifies. In arbitrary games, the manifolds are expected to be complex structures and, as the Reviewer correctly points out, it will not make sense to argue about their monotonicity. To avoid any confusion, the Reviewer's suggestion is well-taken and we removed the possibly misleading statement.

---

### Official Review · Reviewer_jma7 · 2022-10-31

**Confidence:** 2
**Correctness:** 4
**Technical Novelty And Significance:** 2
**Empirical Novelty And Significance:** Not applicable
**Recommendation:** 5

**Clarity, Quality, Novelty And Reproducibility:**

This paper is well-written in general. However I believe there should be more discussion on comparison with PP16.

**Strength And Weaknesses:**

Strengths:

The results are solid. The motivation and proof ideas are well explained.

Weaknesses:

I'm not familiar with this specific topic studied in this paper, however I found it's a follow-up of (and resembles) the EC paper "Average Case Performance of Replicator Dynamics in Potential Games via Computing Regions of Attraction" (PP16) which this paper doesn't seem to give enough credit to. The studies of convergence of RD to NE, region of attraction and average price of anarchy already appeared in PP16, yet this paper doesn't have enough discussion on PP16 at all. PP16 were only mentioned and cited a few places in this paper just like other much less relevant papers, which might cause misleading judgement on the novelty of the results.

The significance of results seems questionable. Theorem 3.2 is a generalization of PP16 to QRD whose proof idea seems similar. The applications Theorems 4.4/4.6 are limited to the 2X2 case.

Some terms are not defined rigorously. For example, $x_k^*$ is a point in simplex in Definition 2.1, what do you mean by a best response is contained in a point?

**Summary Of The Paper:**

This paper proves pointwise convergence of q-replicator dynamics to NE and corresponding bounds on average price of anarchy, generalizing previous works.

**Summary Of The Review:**

I tend to reject due to lack of comparison with PP16 and seemingly incremental results.

After discussion with other reviewers, I'm satisfied with added discussion with PP16 and will raise the score accordingly.

---

> ### Author Response · Authors · 2022-11-16
> **Re: Official Review of Paper6539 by Reviewer jma7**
>
> We thank the reviewer for their feedback. As far as the statement of the reviewer that we haven't given enough credit to PP16 is concerned, please see the updated version of our submission (alterations in blue) in which we have added a paragraph explaining the similarities and the differences with PP16. We also elaborate below. It is true that PP16 is the main precursor of our work. However, our results expand the PP16 paper in multiple ways as follows:
> * The PP16 paper focuses exclusively on quantifying and bounding the APoA of the Replicator Dynamics, in a more restrictive class of 2x2 games; hence the tighter bound.
> * We aim to expand the applicability of the average performance metrics in larger settings with our work. Our first theorem succeeds in this expansion by stating that metrics are well-defined in arbitrarily large potential games; with the exception of a set of them that have measure-zero. This allows for statistical methods for the analysis of larger settings.
> * The 2x2 cases we analyze include more general setups compared to the 2x2 setup in the PP16 paper. They are complex enough that traditional metrics such as the PoA can be proved to be unbounded (see Example 4.1). We are able to prove tight constant bounds for the APoA in these more general settings.
> * To the best of our knowledge, we are the first paper to compare the performance of two optimal no-regret Algorithms.
>
> Given that we are first paper to be able to expand upon PP16 in more than six years I think this showcase that our results are far from trivial. In fact, we hope that the reviewer agrees that our mathematical analysis is rather involved and spans a wide variety of ideas (e.g. Lyapunov functions, invariant functions, stable/unstable manifolds, etc). This is a necessary step in the direction of proving positive APoA bounds in larger instances, which as we show experimentally once again seem to correspond to small constants even in much more complex settings.
>
> On a minor point, Definition 2.1 has been reworded.

---

### Decision · Program_Chairs · 2023-01-20

**Decision:**

Reject

**Justification For Why Not Higher Score:**

As I have detailed in my review, the slightly limited scope of the paper's result were the reason for the score.

**Justification For Why Not Lower Score:**

N/A

**Metareview: Summary, Strengths And Weaknesses:**

The paper considers the problem of quantifying the quality of equilibria a class of dynamics converge to in games. Their contributions are
1. They first show that for a general class of games Q replicator dynamics (which cover GD and some other natural dynamics) converge point wise to Nash Equilibria.
2. They further focus on the quality of the equilibria by studying theoretically and experimentally average metrics (such as average price of anarchy) of the equilibria the dynamics converge to. This latter result is primarily established theoretically for 2x2 symmetric games.

Overall the reviewers had a divided opinion about the paper. While the reviewers unanimously agreed on the technical contribution and the main result which the reviewers agreed is a hard problem to solve and found the solution valuable. On the other hand the reviewers had concerns that the results were somewhat limited due to them being for the specific 2x2 case and the main theorem in that setting also particularly was focussed on specific values. Reviewers also suggested that the initial writing almost seemingly made the reader believe that the question itself was being introduced by the paper.

Overall there is one clear contribution by the paper, it was not fully clear to the reviewers what the contribution to the ICLR community at large might be. The paper is truly borderline taking everything into consideration according to the reviewers and due to the slightly limited scope of the eventual results of the paper, I am making my recommendation. I would like to re-stress that the reviewers appreciated the results' technicality.

**Summary Of Ac-Reviewer Meeting:**

The reviewers met and we discussed the paper in detail. As mentioned in my meta review the reviewers were very divided about the paper. The contributions were very clear to the reviewers as were the limitations and their final scores eventually hinged on the technical contribution vs the limited scope.